# A New Design Problem in the Formulation of a Special Moment Resisting Connection Device for Preventing Local Buckling

**Salvatore Benfratello** 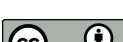**, Luigi Palizzolo * and Santo Vazzano**

Department of Engineering, University of Palermo, Viale delle Scienze Ed. 8, 90128 Palermo, Italy;
salvatore.benfratello@unipa.it (S.B.); santo.vazzano@unipa.it (S.V.)
* Correspondence: luigi.palizzolo@unipa.it (L.P.)

**Abstract:** In the present paper an improved formulation devoted to the optimal design problem of a special moment resisting connection device for steel frames is proposed. This innovative device is called a Limited Resistance Plastic Device (LRPD) and it has been recently proposed and patented by some of the authors. It is thought to be preferably located at the extremes of the beam, connecting the beam end cross section with the relevant column. The typical device is a steel element characterized by symmetry with respect to three orthogonal barycentric planes and constituted by a sequence of three portions with abrupt cross section changes. The main novelty of the present proposal is related to the design of special geometry for the optimal device ensuring that it possesses a reduced resistance with respect to the relevant connected beam element, is characterized by an equivalent bending stiffness equal to the one of the connected beam elements and exhibits full plastic deformations avoiding any local instability phenomenon. The optimal design is formulated as a minimum volume one and is subjected to suitable constraints on the geometry of the device and on its elastic and plastic behavior. The optimization problem is a strongly non-linear programming one and it is solved by adopting an interior-point algorithm that is available in the MATLAB Optimization Toolbox. The numerical simulations are devoted to the most used standard steel profiles (IPE, HE) and the results prove the great reliability of the proposed device. In addition, the relevant elastic and plastic domains of the designed devices are defined, and the expected behavior of the device is verified by appropriate 3D finite element models in the ABAQUS environment.

**Keywords:** moment resisting connections; full plastic deformations; minimum volume design; finite element models; steel design

## 1. Introduction

In many countries, especially in Europe, both civil and industrial construction activity is mainly devoted to the maintenance and restoration of existing constructions. This trend is more intense in those countries rich in ancient and monumental buildings which often require adjustment interventions due to functional recovery and/or to structural security reasons related to standard upgrades.

In this framework the use of steel structures becomes fundamental, due to their strength, lightness, dimensional variety and adaptability to uncommon shapes which make them as ideal solution to perform the requested mechanical, functional and aesthetic purposes.

Clearly, a fundamental aspect is related to the possibility of managing the interference of steel structures with existing ones in the broadest way; that is, to design structures with both optimal strength and stiffness characteristics. In order to obtain the latter optimal characteristics, it is essential that they can be chosen independently of each other while, usually, such independent choice cannot be performed.

In the present paper reference will be made to the most usual steel structures, i.e., the moment resisting frames (MRFs). From a mechanical point of view and specifically based on the seismic resistance model, MRFs are characterized by the onset of plastic hinges at the ends of beams and column bases resulting in an energy dissipation capacity greater

than that available in shear wall systems and braced frames. Many issues are related to the formation of plastic hinges in MRF and to the linked structural consequences; for a discussion on these issues see, e.g., [1–4].

A very important aspect to be considered in MRFs is the problem of connection performance and many papers address this topic [5–10] starting from the extensive damages to the beam-to-column connections of the MRFs occurred during the 1994 Northridge and 1995 Kobe earthquakes. Basically, these studies resulted in the development of pre-qualified connections for use in seismic areas [11], which includes the reduced beam section (RBS) connections. In fact, the connection between beam and column must ensure the occurring of plastic curvature in the beam extreme, preventing local buckling phenomena and brittle rupture of the connection itself. A very large bibliography is available in literature regarding the connections between beams and columns and related issues, see, e.g., [12–20].

The plastic curvature capacity of the involved cross section preventing local buckling phenomena can be evaluated making reference to the cross-section classification reported in the international standards. The role of cross section classification is to identify the extent to which the resistance and rotation capacity of cross sections is limited by its local buckling resistance. To properly define such a limit, the international standards [21–24] define four different classes. Class 1: ductile cross-sections are those possessing rotation capacity able to develop plastic hinges and wide plastic rotation capacity; Class 2: compact cross-sections are those that can reach their plastic moment resistance but show limited plastic rotation capacity; Class 3: semi-compact cross-sections are those that can reach just their limit elastic moment resistance without any plastic rotation; Class 4: slender cross-sections are those in which the buckling phenomena strongly influence the limit elastic moment resistance value. Consequently, plastic analysis can be conducted only for structural members whose cross-sections belong to Class 1 or 2.

Furthermore, in order to avoid undesired brittle rupture of the beam-column connection, suitable reduced beam section technology can be utilized [12–20]. Unfortunately, such intervention causes a related reduction of the stiffness of the involved beam element, contrary to the previously fixed requirement, i.e., to design structures with both optimal strength and stiffness characteristics.

To overcome these disadvantages, in the recent past, the authors proposed a new connection device realizing a special moment resisting connection for steel elements called the Limited Resistance Plastic Device (LRPD) [25–31]. The LRPD satisfies the requirements, ensuring the independence between strength and stiffness and, as a consequence, allowing an optimal mechanical performance for the structure that is suitably equipped. In the referenced papers, different versions of the connection have been presented, starting from the first device representing a simple rigid perfectly plastic hinge described by a concentrated plasticity model up to the most recent advanced version modeled as a distributed plasticity device. In the same studies no constraints have been imposed on the class classification of the device cross sections and reference has been made to the plastic behavior described by approximate function, as reported in the most common international standards.

The aim of the present paper is to propose an improvement of the LRPD that is related to its new geometry, to the new design problem formulation and to the more rigorous description of its real plastic behavior. In particular, the development of full plastic curvature in the device cross sections, avoiding any local buckling phenomenon, is obtained by imposing in the new improved optimal design problem that the characteristic cross section of the device appertains to Class 1. For the optimal device characterized by improved geometry, the plastic behavior has been described by referring to the real domains represented by nonlinear functions of axial force and bending moment. Furthermore, a wide FEM of numerical investigations to check the ability of the LRPD of developing plastic deformations is performed, confirming the good consistency of the theoretical position of the problem and the wide reliability of the device.

## 2. Geometrical and Mechanical Characteristics of LRPD

The device studied in the present paper is an evolution of that reported in other foregoing papers (see, e.g., [25–31]) and consists of a steel element with suitably assigned features aimed to substitute a portion of a given standard I-shaped steel profile. In Figure 1, the typical standard steel profile is sketched and its geometry is reported in Table 1.

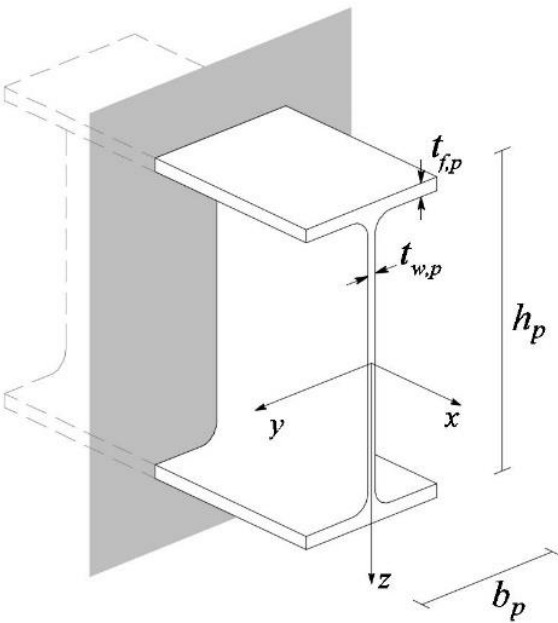

**Figure 1.** Typical standard I-shaped steel profile.

**Table 1.** Geometrical characteristics of the standard steel profile.

| Adopted Symbol | Description |
| :---: | :---: |
| $b_p$ | flange width |
| $h_p$ | total height |
| $t_{w,p}$ | web thickness |
| $t_{f,p}$ | flange thickness |

In Figure 2, a scheme of the device is reported. As it is possible to observe, the overall device is assumed to be inscribed in a parallelepiped of dimensions $\ell \times b_p \times h_p$.

The device is made up by three different parts: two outers (equal each other) and one inner. From a geometrical point of view, the device possesses the following geometrical features: (i) it is characterized by symmetry with respect to three orthogonal barycentric planes; (ii) the thickness of flanges of the two outer portions is greater than that of the inner portion; (iii) the flanges of all portions possess a unique common medium plane; (iv) the thickness of web is the same in all the portions. It is important to emphasize that the actual version is an evolution of previous models (see, e.g., [31]) and, in particular, the main novelties are: (a) the flange width $b_i$ of the inner part is assumed as different with respect to that of the outer portions; (b) the thickness of web is the same in all the portions.

For a complete geometrical representation of the device, in Figure 3 the sketch of both the outer and inner portions, as well as a lateral view of the device, are reported while in Table 2 the adopted symbols are described.

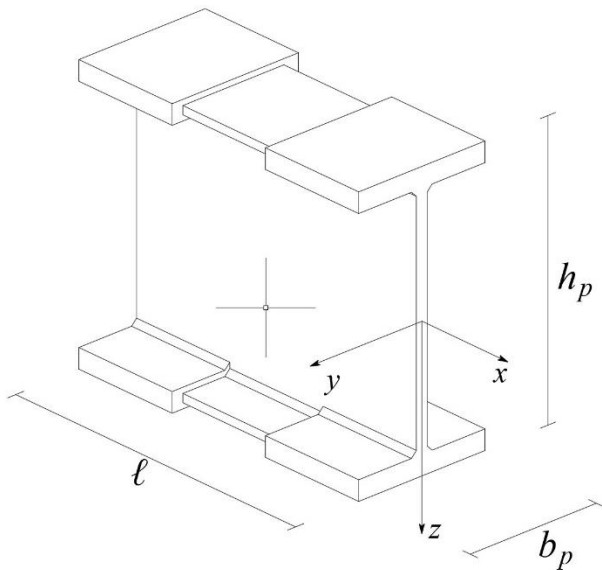

**Figure 2.** Device scheme.

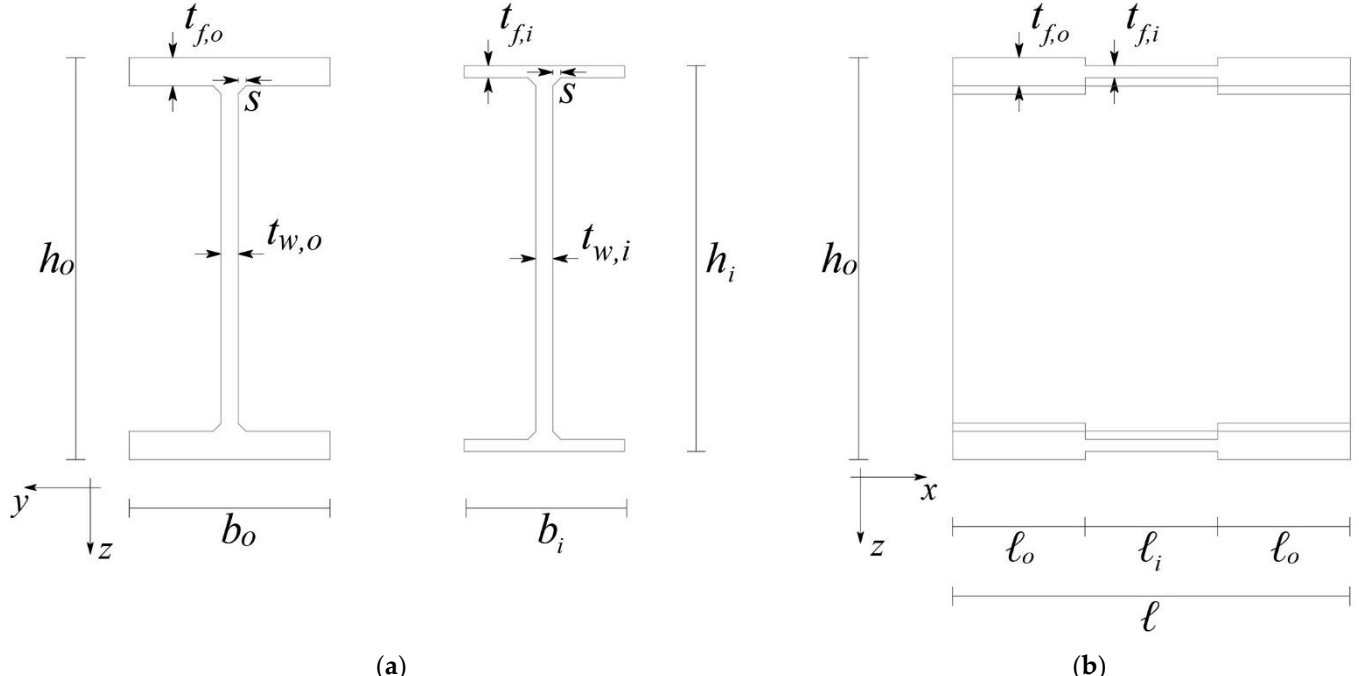

| (**a**) | (**b**) |

**Figure 3.** Device geometry: (**a**) cross sections of the inner and outer portions; (**b**) lateral view.

It is worth noting that the imposed overall geometry for the device implies that: $b_o = b_p$, $h_o = h_p$ (see Figure 2). In addition, as a further novelty, in the present new device the web thickness is constant for all the portions and equal to one of the original standard steel I-shaped profiles: i.e., $t_{w,o} = t_{w,i} = t_{w,p}$. From a mechanical point of view, the connection with the structural elements is generally thought of as a bolted plate and back-plate system, hereafter assumed as a perfect rigid joint, as reported in Figure 4, where bolts are not shown for the sake of simplicity. The analysis of the mechanical behavior of this connection, together with its influence of the overall behavior of the device, is not faced in the present paper and it will be treated in future developments.

**Table 2.** Geometrical characteristics of the device.

| Adopted Symbol | Description |
|:---:|:---:|
| Outer portions | |
| $b_o$ | flange width |
| $h_o$ | total height |
| $t_{w,o}$ | web thickness |
| $t_{f,o}$ | flange thickness |
| $\ell_o$ | length |
| Inner portion | |
| $b_i$ | flange width |
| $h_i$ | total height |
| $t_{w,i}$ | web thickness |
| $t_{f,i}$ | flange thickness |
| $\ell_i$ | length |
| Overall device | |
| $\ell$ | total length |
| $s$ | welding size |

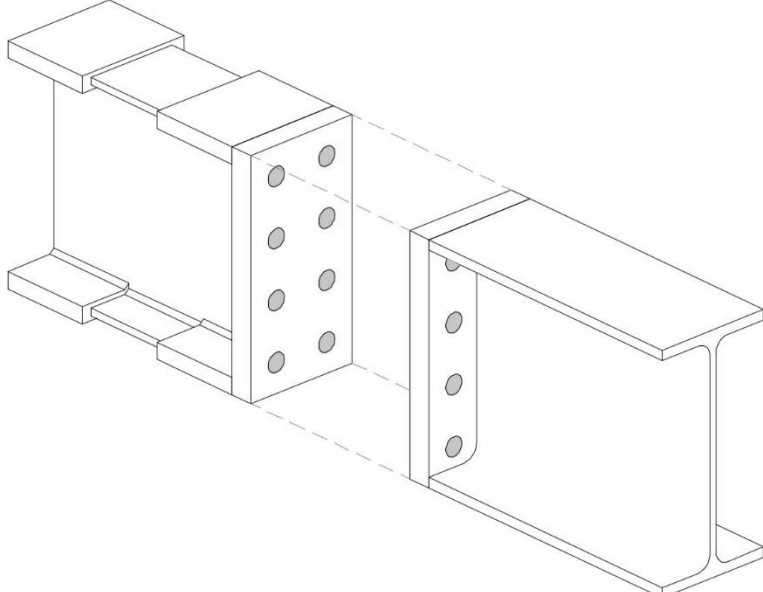

**Figure 4.** 3D view of the device and of the connection with the beam.

Among the geometrical characteristics of the device reported above, the welding size *s* has been introduced (Figure 3). In fact, from a technological point of view, the device is thought of as being obtained by the welding of steel plates with suitable thicknesses (for further details see [31]).

For the developments of the paper, it is necessary to define the cross-section area $A$, the moment of inertia $I$, the elastic resistance modulus $W_{el}$ and the plastic resistance modulus $W_{pl}$, respectively, for each portion of the device. These quantities can be easily derived from basic geometry definitions and are reported in detail in [30,31].

Plane frames analysis will be adopted in this paper; therefore, the constitutive elements are generally subjected to an axial force, shear force and bending moment. By neglecting the influence of the shear force, as for flexural systems, the limit elastic and plastic behavior of the device will be described, making reference just to the axial force and bending moment.

Respecting the contents of the most recent international structural standards (see, e.g., [21–24]), referring as previously indicated just to I-shaped Class 1 cross-sections,

the boundary of the elastic domain (see Figure 5) is simply described by the following dimensionless conditions:

$$\left|\frac{N}{N_{el}} + \frac{M}{M_{el}}\right| = 1, \quad \left|\frac{N}{N_{el}} - \frac{M}{M_{el}}\right| = 1 \tag{1}$$

where $N_{el} = A\,f_y$ is the elastic limit value of the axial force (coincident with the plastic limit one) and $M_{el} = W_{el}\,f_y$ the elastic limit value of the bending moment, with $f_y$ the material yield stress.

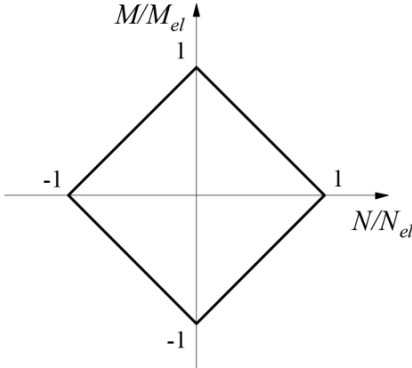

**Figure 5.** Dimensionless elastic domain.

The boundary of the plastic domain, always related to I-shaped Class 1 cross-sections is reported in Figure 6 and, always according with the referenced international structural standards, is described by the following dimensionless equations:

$$\left|\frac{N}{N_{pl}} + (1 - 0.5a)\frac{M}{M_{pl}}\right| = 1, \quad \left|\frac{N}{N_{pl}} - (1 - 0.5a)\frac{M}{M_{pl}}\right| = 1 \tag{2}$$

$$\left|\frac{M}{M_{pl}}\right| = 1 \tag{3}$$

where $N_{pl} = A\,f_y$ is the plastic limit value of the axial force, $M_{pl} = W_{pl}\,f_y$ is the plastic limit value of the bending moment and where the parameter $a$ is defined as follows:

$$a = \left(A - 2b_p t_{f,p}\right)/A \le 0.5 \tag{4}$$

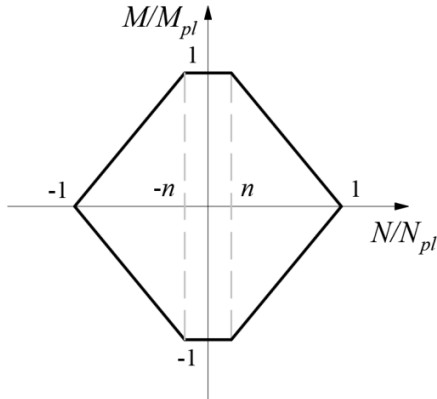

**Figure 6.** Dimensionless yield domain: the value of $n$ depends on the cross-section profile ($n = 0.5a$).

It is worth noting that the size of the plateau of the domain in Figure 6 is 2*n*, and, utilizing Equations (2) and (3), it can be deduced.

$$n = \frac{N}{N_{pl}} = 0.5a \tag{5}$$

Due to the imposed constraint on parameter *a*, it always results $n = N/N_{pl} \leq 0.25$, according to the referenced standards.

The dimensionless yield domain sketched in Figure 6, and also reported in the international standards, represents a good reference from a practical point of view even if the outline of such a domain is linearized and approximated. The real yield domain boundary is nonlinear and it can be drawn, as described in the following.

As it can be observed by referring to Figure 7, where the typical yield domain of an I-shaped cross section is sketched in the *N*, *M* plane, its boundary is symmetric with respect to the coordinate axes and, as a consequence, for its analytical determination reference can be made just to the first quarter of the plane.

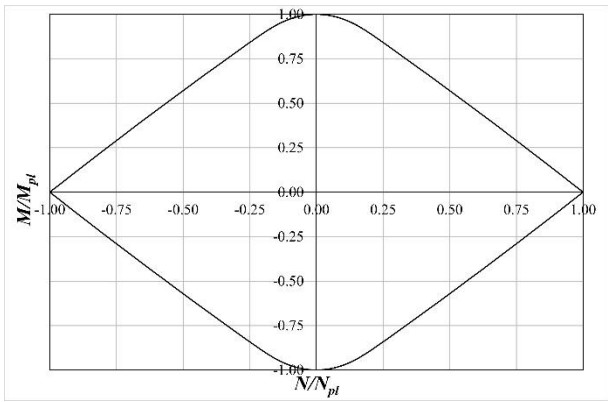

**Figure 7.** Dimensionless yield domain for beam with H cross-section profile.

For each position of the neutral axis $-\frac{h_p}{2} \leq z_n \leq \frac{h_p}{2}$ (see Figure 8), the entire cross section is assumed to be plasticized and the related values for the yield axial force and yield bending moment read:

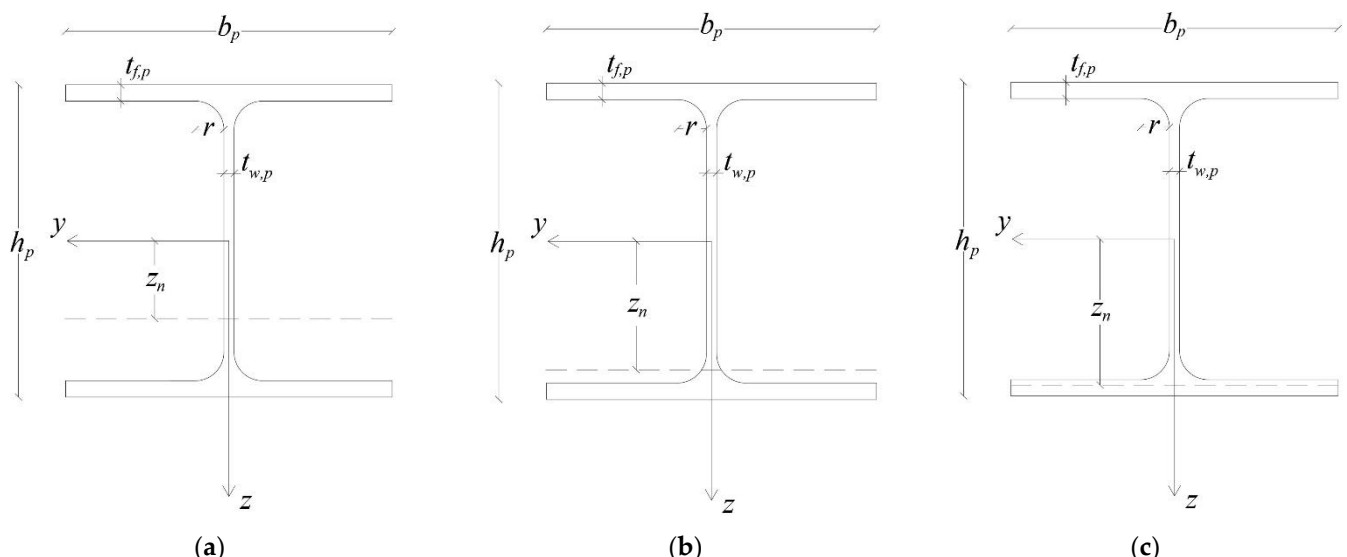

**Figure 8.** Standard profile geometry for yield domain: (**a**) neutral axis crossing the web; (**b**) neutral axis crossing the connection between web and flange; (**c**) neutral axis crossing the flange.

for $0 \leq z_N \leq \frac{h_p}{2} - t_{f,p} - r$ (Figure 8a)

$$
\begin{aligned}
N &= 2 \, t_{w,p} \, z_n \, f_y; \\
M &= M_{pl} - t_{w,p} \, z_n^2 \, f_y
\end{aligned}
\tag{6}
$$

for $\frac{h_p}{2} - t_{f,p} - r \leq z_n \leq \frac{h_p}{2} - t_{f,p}$ (Figure 8b)

$$
N = f_y \left[ A - 2b_p t_{f,p} - 2t_{w,p}\left(\frac{h_p}{2} - t_{f,p} - z_n\right) - 4F_1 \right]
$$

$$
M = f_y \left\{ b_p t_{f,p}\left(h_p - t_{f,p}\right) + t_{w,p}\left[ \left(\frac{h_p}{2} - t_{f,p}\right)^2 - z_n^2 \right] + 4F_1(z_n + F_2) \right\}
\tag{7}
$$

where

$$
F_1 = r\left(\frac{h_p}{2} - t_{f,p} - z_n\right) - \frac{r^2}{4}(\vartheta - sen \, \vartheta)
\tag{8}
$$

$$
\vartheta = 2 \, arccos\left( \frac{z_n + t_{f,p} + r - \frac{h_p}{2}}{r} \right)
\tag{9}
$$

$$
F_2 = \frac{F_3}{F_1}
\tag{10}
$$

$$
F_3 = \frac{r}{2}\left(\frac{h_p}{2} - t_{f,p} - z_n\right)^2 - \frac{r^2}{4}(\vartheta - sen \, \vartheta)\left[ \frac{r^2 sen \, \vartheta\left(\frac{h_p}{2} - t_{f,p} - z_n - r\right) + 2 \, r^3 sen \frac{\vartheta}{2}}{\frac{3}{2}r^2(\vartheta - sen \, \vartheta)} + \frac{h_p}{2} - t_{f,p} - z_n - r \right]
\tag{11}
$$

for $\frac{h_p}{2} - t_{f,p} \leq z_n \leq \frac{h_p}{2}$ (Figure 8c)

$$
N = f_y \left[ A - 2b_p\left(\frac{h_p}{2} - z_n\right) \right];
\tag{12}
$$

$$
M = b_p f_y \left( \frac{h_p^2}{4} - z_n^2 \right)
\tag{13}
$$

Now, referring to the proposed device, analogous yield domains can be defined and, taking into account the special requirements, only the plastic behavior of the inner portion must be checked. Furthermore, neglecting, as is usual in technical applications, the mechanical welding contribution, the yield domain boundary can be obtained as the envelope of the following functions (effective in the range $-N_{pl} \leq N \leq N_{pl}$ and $-M_{pl} \leq M \leq M_{pl}$):

for $0 \leq z_N \leq \frac{h_i}{2} - t_{f,i}$

$$
M = \left( W_{pl,i} - t_{w,p} z_n^2 \right) f_y = M_{pl,i} - \frac{N^2}{4t_{w,p}f_y}
\tag{14}
$$

for $\frac{h}{2} - t_f \leq z_n \leq \frac{h}{2}$

$$
M = f_y \, b_i \left[ \frac{\left(h^* + t_{f,i}\right)^2}{4} - \left( \frac{N - f_y A_i}{2b_i f_y} + \frac{h^* + t_{f,i}}{2} \right)^2 \right]
\tag{15}
$$

These functions, rewritten in suitable form, will be adopted as bounds on the LRPD resistance in the optimal design problem which will be presented in the following section.

## 3. Optimal Design Formulation

The present section is devoted to the formulation of a new improved version of the optimal design problem for the proposed device. The essential novelty of the formulation regards a reduced and more compact form of the design variable vector and the constraint on the desired Class of the cross section of the inner portion of the device. As previously reported, the LRPD is constituted by an inner portion of length $\ell_i$, with geometrical I-shaped cross section features reported in Section 2, and by two symmetrically placed outer portions, both of length $\ell_o$ and equal I- shaped cross section with geometrical features as described in Section 2. The device volume is chosen as the objective function to be minimized.

It is worth noting that, to ensure the occurrence of appropriate plastic deformation fields in the inner portion, the length of the inner part $\ell_i$ must satisfy a suitable lower bound [28]. In the referenced paper, the length $\ell_i$ is evaluated as a linear function of the height of the original beam cross section introducing an appropriate scalar factor $\beta$: $\ell_i = \beta h_p$ where $0.5 \leq \beta \leq 1$ is suggested.

In Figure 9 all the geometrical characteristics of the device are indicated and, in particular, the assumed design variables are highlighted in red. In the referenced Figure, as previously stated, $h_o = h_p$, $b_o = b_p$ and $t_{w,o} = t_{w,i} = t_{w,p}$ have been assumed. Finally, the distance between the medium planes of the flanges $h^*$ is considered as a relevant variable:

$$h^* = h_o - t_{f,o} = h_i - t_{f,i} \tag{16}$$

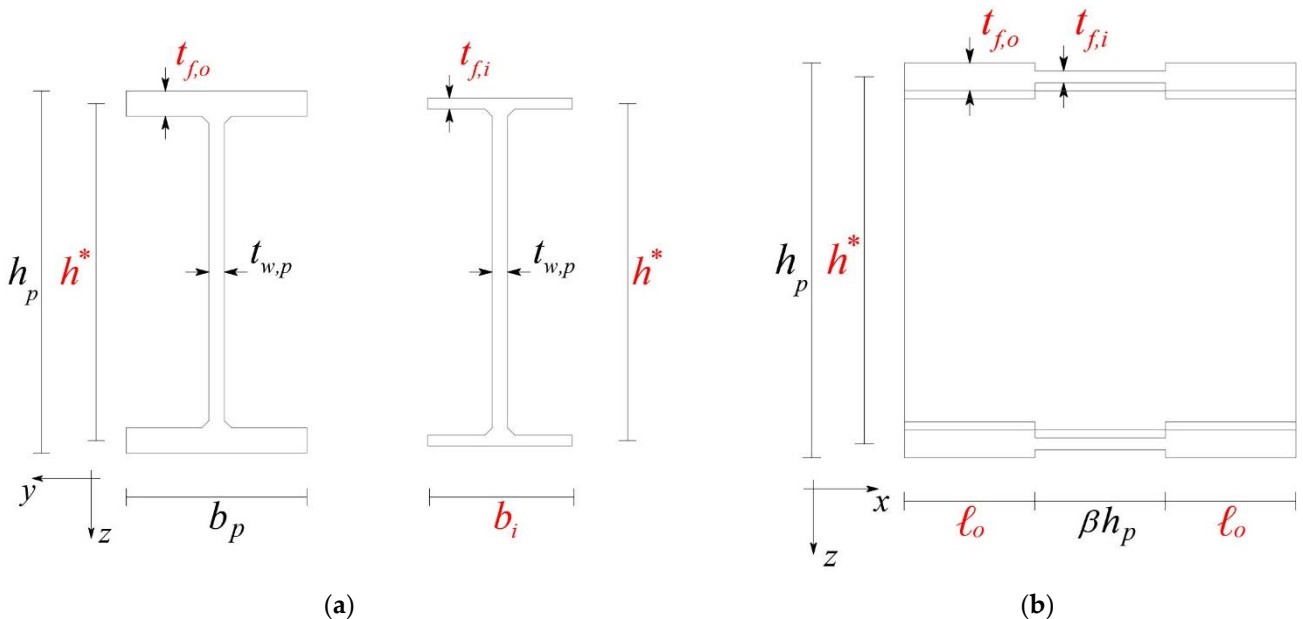

**Figure 9.** Device geometry and design variables: (**a**) cross sections of the inner and outer portions; (**b**) lateral view.

As a consequence, the design variable vector is the following one:

$$d^T = \left| h^* \ b_i \ t_{f,i} \ t_{f,o} \ \ell_o \right| \tag{17}$$

The objective function to be minimized is the following:

$$
\begin{aligned}
f(d) = A_i \ell_i + 2 A_o \ell_o &= \left[ 2 b_i t_{f,i} + t_{w,p} \left( h^* - t_{f,i} \right) \right] \ell_i + 2 \left[ 2 b_p t_{f,o} + t_{w,p} \left( h^* - t_{f,o} \right) \right] \ell_o \\
&= \left[ 2\, b_i\, t_{f,i} + t_{w,p} \left( h^* - t_{f,i} \right) \right] \beta h_p + \left[ 4 b_p t_{f,o} + 2 t_{w,p} \left( h^* - t_{f,o} \right) \right] \ell_o
\end{aligned} \tag{18}
$$

i.e., the volume of the searched device.

The design variables must satisfy appropriately selected bounds, reported in the following relations:

$$0 \le h^* \le h_p - t_{f,p} \tag{19}$$

$$3t_{w,p} \le b_i \le b_p \tag{20}$$

$$0 \le t_{f,i} \le t_{f,p} \tag{21}$$

$$t_{f,p} \le t_{f,o} \le {h_p}/{2} \tag{22}$$

$$0 \le \ell_o \le \infty \tag{23}$$

where $t_{f,p}$, as already defined in Section 2, is the flange thickness of the original beam element cross section.

Furthermore, the device geometry implicates:

$$h^* + t_{f,o} = h_p \tag{24}$$

If the I-shaped cross section of the inner portion is imposed to belong to the relevant Class 1, the following relation must be satisfied (see [21,24]).

$$\frac{b_i - 3t_{w,p}}{2t_{f,i}} \le 9\varepsilon = 9\sqrt{\frac{235}{f_y}} \qquad \rightarrow \qquad b_i - 18t_{f,i}\sqrt{\frac{235}{f_y}} \le 3t_{w,p} \tag{25}$$

In addition, the device is designed to be in a limit resistance condition for assigned couples of values of the axial force $N_a$ and of the bending moment $M_a$. Therefore, the following mechanical condition must be respected:

$$b_i\, t_{f,i} h^* + t_{w,p} \left(\frac{h^* - t_{f,i}}{2}\right)^2 = \frac{N_a^2}{4\, t_{w,p}\, f_y{}^2} + \frac{M_a}{f_y} \tag{26}$$

if the neutral axis crosses through the web cross section, or

$$\frac{N_a}{2}\left[2t_{f,i} + \frac{t_{w,p}}{b_i}\left(h^* - t_{f,i}\right) - \frac{N_a}{2b_i\, f_y} - \left(h^* + t_{f,i}\right)\right] + f_y\left[b_i t_{f,i} + \frac{t_{w,p}}{2}\left(h^* - t_{f,i}\right)\right]\left[\left(h^* + t_{f,i}\right) - t_{f,i} - \frac{t_{w,p}\left(h^* - t_{f,i}\right)}{2b_i}\right] = M_a \tag{27}$$

if the neutral axis crosses through one of the flanges of the cross section.

Finally, the equivalent bending stiffness of the device with total length $\ell = \ell_i + 2\ell_o$ must be not less than the one characterizing a portion of the original beam element with the same length. The following inequality must be satisfied (see [26]):

$$\frac{\ell_i}{\ell_o} - 2\frac{I_i}{I_o}\left(\frac{I_o - I_p}{I_p - I_i}\right) \le 0 \tag{28}$$

that, written as function of the design variables, reads:

$$\frac{\beta h_p}{\ell_o} - \frac{4b_i\, t_{f,i}^3 + 12b_i t_{f,i} h^{*2} + 2t_{w,p}\left(h^* - t_{f,i}\right)^3}{2b_p\, t_{f,o}^3 + 6b_p t_{f,o} h^{*2} + t_{w,p}\left(h^* - t_{f,o}\right)^3}\left[\frac{2b_p\, t_{f,o}^3 + 6b_p t_{f,o} h^{*2} + t_{w,p}\left(h^* - t_{f,o}\right)^3 - 12I_p}{12I_p - 2b_i\, t_{f,i}^3 - 6b_i t_{f,i} h^{*2} - t_{w,p}\left(h^* - t_{f,i}\right)^3}\right] \le 0 \tag{29}$$

where $I_p$ is the moment of inertia of the original beam element.

The optimal design problem can be formulated as follows:

$$\min_{(d)} f(d) \tag{30}$$

Subjected to:

$$d_{low} \le d \le d_{upp} \tag{31}$$

$$A_{eq}d = b_{eq} \tag{32}$$

$$A_{in}d = b_{in} \tag{33}$$

$$G_{eq}(d) = q_{eq} \tag{34}$$

$$G_{in}(d) \leq q_{in} \tag{35}$$

In the above reported relations, the adopted scalar functions, vectors and matrices have the following form:

$$d_{low}^T = \begin{vmatrix} 0 & 3t_{w,p} & 0 & t_{f,p} & 0 \end{vmatrix} \tag{36}$$

$$d_{upp}^T = \begin{vmatrix} h_p - t_{f,p} & b_p & t_{f,p} & \dfrac{h_p}{2} & \infty \end{vmatrix} \tag{37}$$

$$A_{eq} = \begin{vmatrix} 1 & 0 & 0 & 1 & 0 \end{vmatrix}; \quad b_{eq} = h_p \tag{38}$$

$$A_{in} = \begin{vmatrix} 0 & 1 & -18\sqrt{\dfrac{235}{f_y}} & 0 & 0 \end{vmatrix}, \qquad b_{in} = 3t_{w,p} \tag{39}$$

$$G_{eq}(d) = b_i\, t_{f,i} h^* + t_{w,p}\left(\dfrac{h^* - t_{f,i}}{2}\right)^2, \quad q_{eq} = \dfrac{N_a^2}{4\, t_{w,p}\, f_y{}^2} + \dfrac{M_a}{f_y} \tag{40}$$

if the neutral axis cuts through the web cross section,

$$G_{eq}(d) = \dfrac{N_a}{2}\left[2t_{f,i} + \dfrac{t_{w,p}}{b_i}\left(h^* - t_{f,i}\right) - \dfrac{N_a}{2b_i\, f_y} - \left(h^* + t_{f,i}\right)\right] + \\ f_y\left[b_i t_{f,i} + \dfrac{t_{w,p}}{2}\left(h^* - t_{f,i}\right)\right]\left[\left(h^* + t_{f,i}\right) - t_{f,i} - \dfrac{t_{w,p}\left(h^* - t_{f,i}\right)}{2b_i}\right] \tag{41}$$

$$q_{eq} = M_a$$

if the neutral axis cuts through one of the flanges cross section,

$$G_{in}(d) = \dfrac{\beta h_p}{\ell_o} - \dfrac{4b_i\, t_{f,i}^3 + 12b_i t_{f,i} h^{*2} + 2t_{w,p}\left(h^* - t_{f,i}\right)^3}{2b_p\, t_{f,o}^3 + 6b_p t_{f,o} h^{*2} + t_{w,p}\left(h^* - t_{f,o}\right)^3} \\ \left(\dfrac{2b_p\, t_{f,o}^3 + 6b_p t_{f,o} h^{*2} + t_{w,p}\left(h^* - t_{f,o}\right)^3 - 12 I_p}{12 I_p - 2b_i\, t_{f,i}^3 - 6b_i t_{f,i} h^{*2} - t_{w,p}\left(h^* - t_{f,i}\right)^3}\right) \tag{42}$$

$$q_{in} = 0$$

The above formulated problem is a non-linear programming one and for its solution a suitable solver is utilized by adopting an interior-point algorithm. In particular, the "fmincon" solver will be implemented in the application stage by utilizing the MATLAB Optimization Toolbox.

## 4. Application

In this section the optimal design for the LRPD described in the foregoing section is applied to the cases of two I-shaped cross section standard steel (S235) profiles. The selected profiles are the IPE400 and HEA300 which represent, in the authors' opinion, two meaningful examples of a typical beam element that is adopted in building constructions.

The first step in applying the proposed optimal design is to select the values of the ultimate axial force and bending moment for the device. In order to do this, the elastic and plastic domains for both the selected profiles have been obtained and reported in Figure 10 by utilizing Equations (1) and (6)–(13).

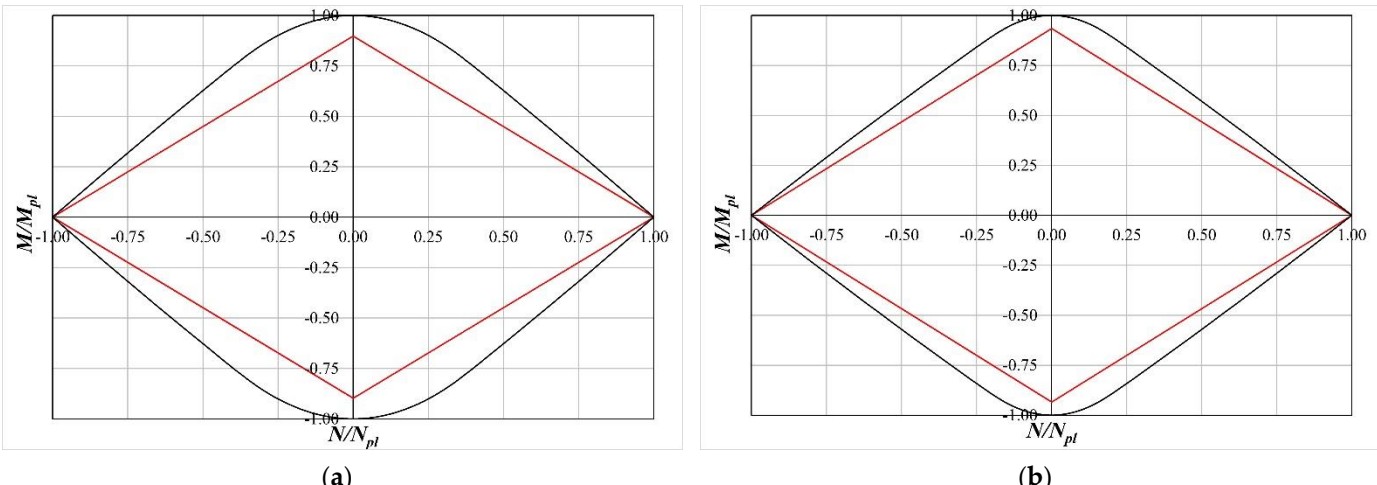

**Figure 10.** Dimensionless elastic (red line) and yield (black line) domains: (**a**) IPE400; (**b**) HEA300.

On the basis of the domains reported in Figure 10, for both the selected commercial profiles two different loading conditions have been selected to perform the optimal design: the first one is characterized by a high bending moment and a small axial force; in the second one the situation is reverted with a high axial force and a relatively small bending moment. Clearly, both situations are such that the mechanical behavior of the standard profile is always elastic. These situations have been referred to as DP1 and DP2 in the following and are reported in Figure 11 together with the elastic and yield domains already reported in the previous figure.

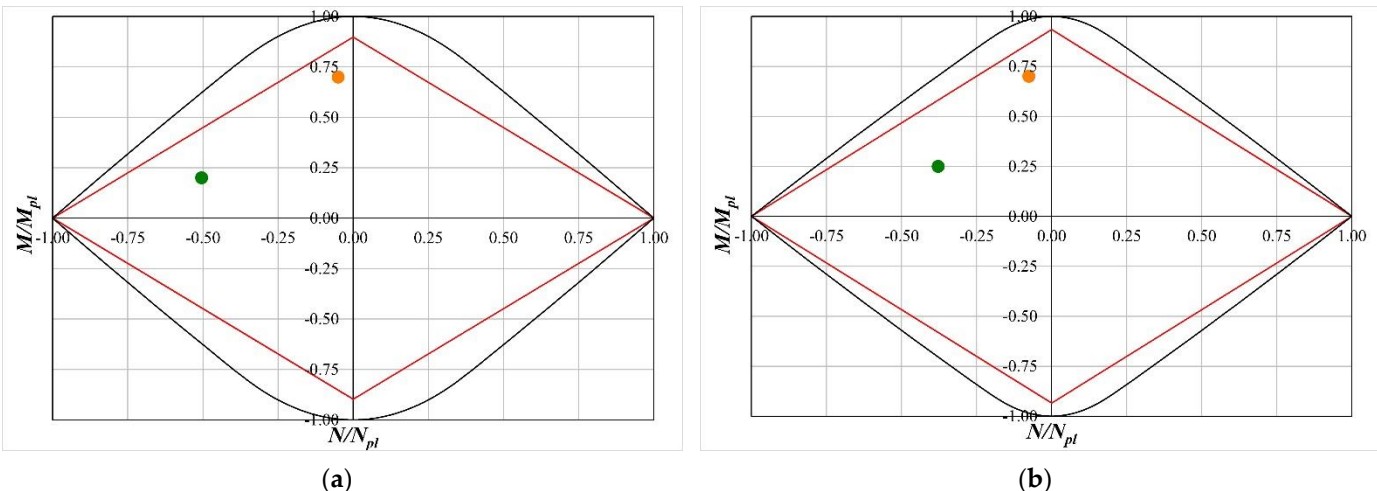

**Figure 11.** Dimensionless elastic (red line) and yield (black line) domains with selected design points (DP1 orange point; DP2 green point): (**a**) IPE400; (**b**) HEA300.

The optimal designs of the devices related to the chosen points (DP1 and DP2) are determined by solving problem (30)–(35). The input values for each problem are reported in Table 3 and in Table 4 the obtained results are reported.

**Table 3.** Input parameters (all dimensions are in mm except for $N_a$, $M_a$, $f_y$ and $I_p$, whose dimensions are kN, kNm, MPa and mm$^4$, respectively).

| | IPE400 | | HEA300 | |
|---|---|---|---|---|
| | **DP1** | **DP2** | **DP1** | **DP2** |
| $b_p$ | 180.0 | 180.0 | 300.0 | 300.0 |
| $h_p$ | 400.0 | 400.0 | 290.0 | 290.0 |
| $t_{w,p}$ | 8.6 | 8.6 | 8.5 | 8.5 |
| $t_{f,p}$ | 13.5 | 13.5 | 14.0 | 14.0 |
| $I_p$ | 231,300,000 | 231,300,000 | 182,600,000 | 182,600,000 |
| $f_y$ | 235.0 | 235.0 | 235.0 | 235.0 |
| $\beta$ | 0.5 | 0.5 | 1.0 | 1.0 |
| $N_a$ | $-100.000$ | $-1000.000$ | $-200.000$ | $-1000.000$ |
| $M_a$ | 215.000 | 61.400 | 227.500 | 81.300 |

**Table 4.** Optimal designs (all dimensions are in mm).

| | IPE400 | | HEA300 | |
|---|---|---|---|---|
| | **DP1** | **DP2** | **DP1** | **DP2** |
| $b_i$ | 130.280 | 165.175 | 233.957 | 219.196 |
| $t_{f,i}$ | 13.500 | 7.743 | 14.000 | 10.761 |
| $\ell_i$ | 200.000 | 200.000 | 290.000 | 290.000 |
| $h^*$ | 368.835 | 368.524 | 262.128 | 261.655 |
| $t_{f,o}$ | 31.165 | 31.476 | 27.872 | 28.345 |
| $\ell_o$ | 120.043 | 208.876 | 179.900 | 362.786 |

As is easy to recognize by an examination of Table 3, the selected value of $\beta = \ell_i/h_p$ is different for IPE and HEA profiles. Specifically, it has been assumed as the lower limit of the suggested interval $\beta = 0.5$ (see [28]) for the IPE profile and the upper one $\beta = 1$ for the HEA profile. This choice is due to the remark that the web flexural contribution for IPE profile is certainly greater than that for HEA profile. Consequently, for the beam with the IPE cross section the plastic deformations easily spread along the web.

To check the reliability of the proposed optimal design, the elastic and yield domains of the inner part of the device have been determined by utilizing Equations (14) and (15). These domains are reported in Figure 12 for the case of the IPE400 profile and in Figure 13 for the case of the HEA300 profile, respectively. An examination of these Figures confirms that the selected design points belong to the yield domain of the inner part of the device designed for that assigned point.

The next step consists of checking the mechanical behavior of the designed devices by means of a suitable FEM analysis performed in an ABAQUS environment. In Figure 14 the model for the LRPD designed for the IPE400 profile and design point 1 is sketched (models for the other LRPDs are similar). The model is constituted by four different parts, sketched in different colors in Figure 14. The central part (sketched in dark green) is the LRPD whose geometrical characteristics have been obtained by the optimal design. The first and third parts (sketched in dark white and dark red, respectively) are equal to each other and the related geometrical characteristics are equal to those of the outer parts of LRPD. The length of these parts is equal to that of the LRPD so that the overall length of the model, without the loading plate, is equal to $3\ell$. The last part (sketched in dark blue) is a plate where the loadings are applied with a thickness of $t = 30$ mm. Each part has been tied to the next one to ensure a perfect kinematic compatibility. The overall model is that of a cantilever beam so that the white part is clamped at its end section while the dark blue part is free. The adopted mechanical behavior for the material is an elastic-perfectly plastic one with Young's modulus equal to 210 GPa, yield stress equal to 235 MPa, ultimate strain equal to 25% and adopted mesh is 5 mm with hexahedral standard 3D stress elements. The loading plate is constituted by linear elastic material with a stiffness that is much greater than the

one of the adopted steel (210 TPa), so that any influence of the pressure distribution is avoided. Two different pressure loads (pink arrows in Figure 14) have been applied to the plate: the first one is uniform and its resultant represents the axial force acting on the device; the second one is a linearly varying one (along *y*-axis) with zero value at barycentric *x*-axis of the plate and its resultant is equal to the bending moment acting on the device. The intensity of each load ranges between zero and the maximum value (corresponding to the design load of LRPD) and this interval has been divided in 100 steps.

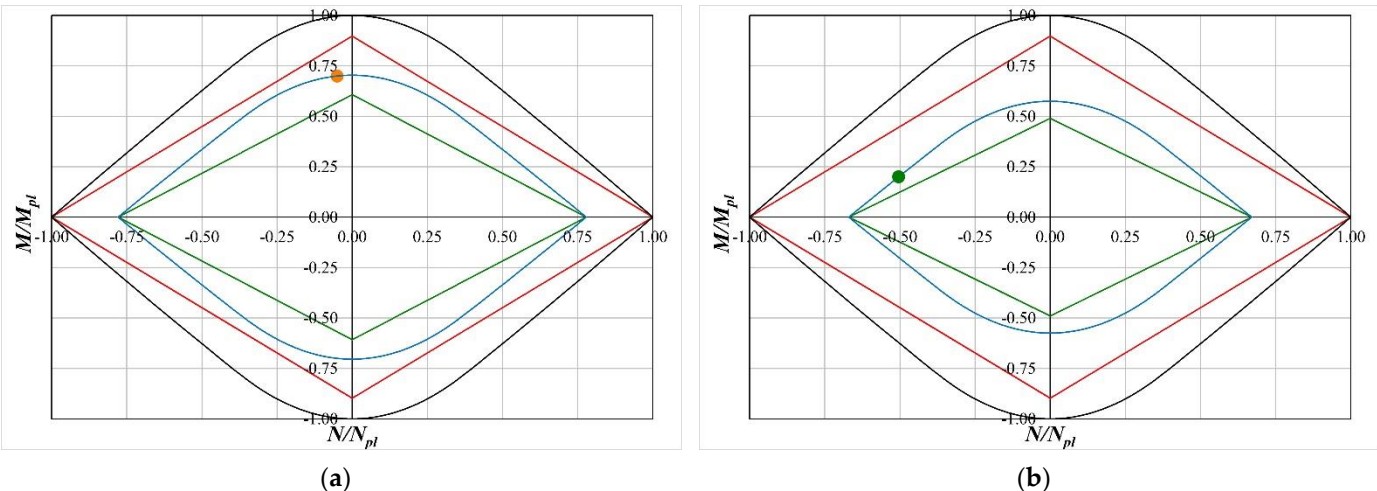

**Figure 12.** Dimensionless domains: IPE400 elastic (red line); IPE400 yield (black line); *LRPD* elastic (green line); *LRPD* yield (blue line). (**a**) design point 1; (**b**) design point 2.

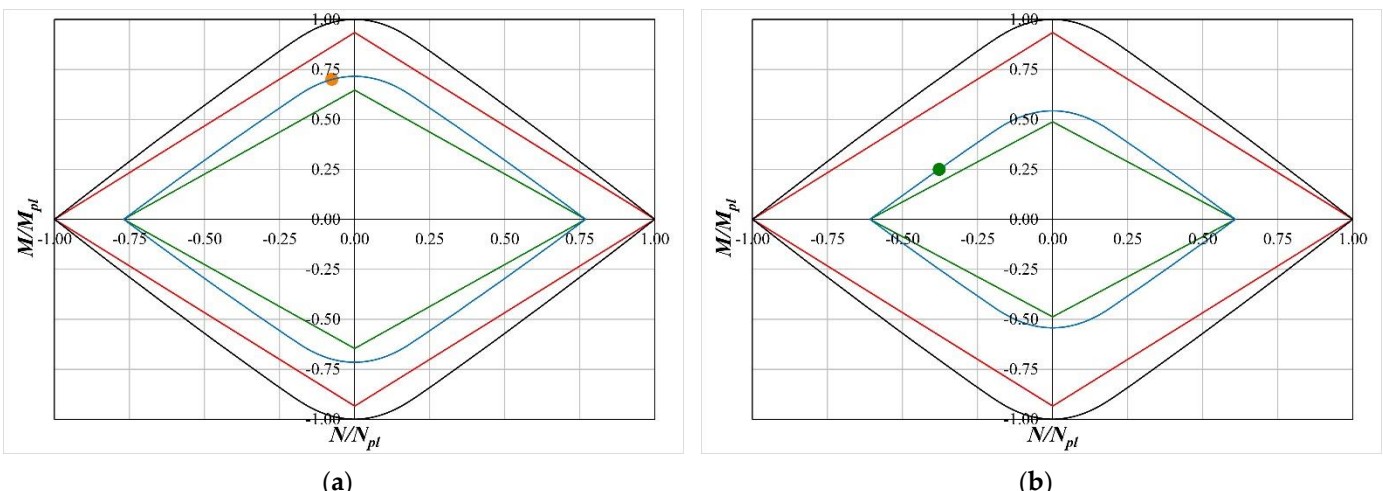

**Figure 13.** Dimensionless domains: HEA300 elastic (red line); HEA300 yield (black line); *LRPD* elastic (green line); *LRPD* yield (blue line). (**a**) design point 1; (**b**) design point 2.

The evaluation of the mechanical behavior of the proposed LRPD has been performed by evaluating the yielding level of the inner part when the loads reach their assigned maximum values. To do this in Figures 15–18, the von Mises' stress maps obtained by FEM analysis are reported, for each profile and for each design point. An examination of these figures immediately reveals that in all the examined cases the inner part of the LRPD is fully plasticized confirming the validity and the effectiveness of the proposed optimal design.

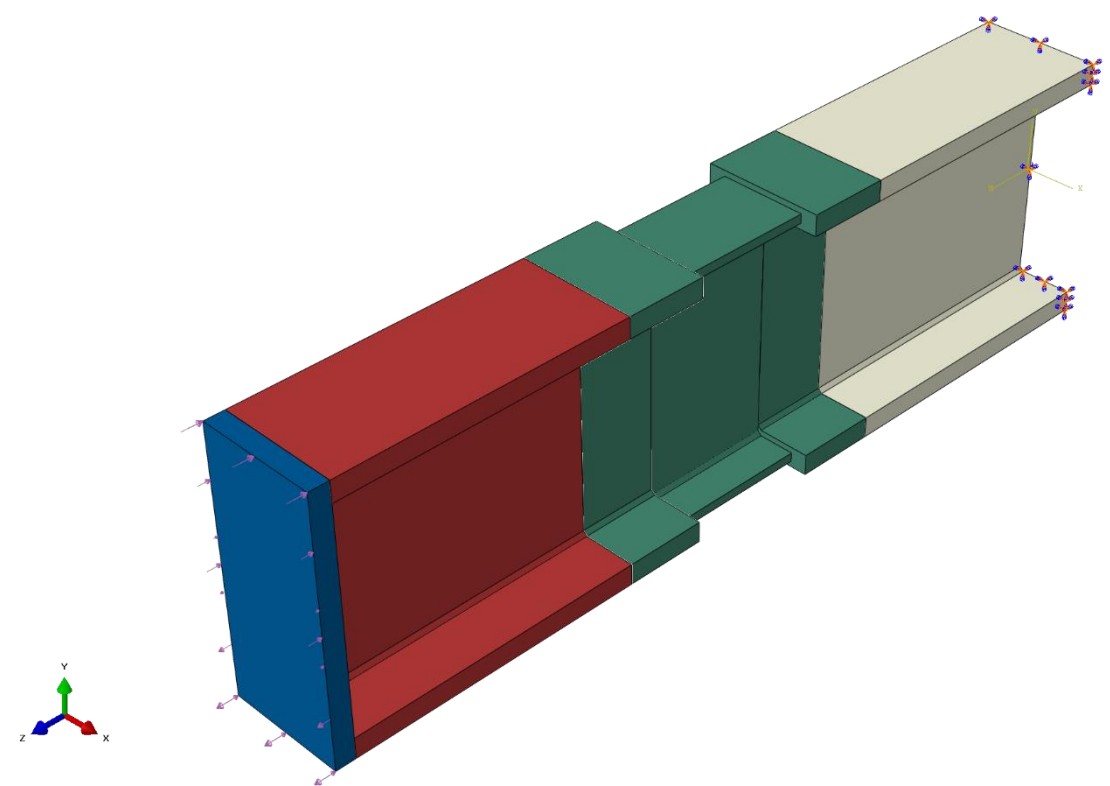

**Figure 14.** ABAQUS model adopted for the numerical simulation of the device related to the IPE 400 profile.

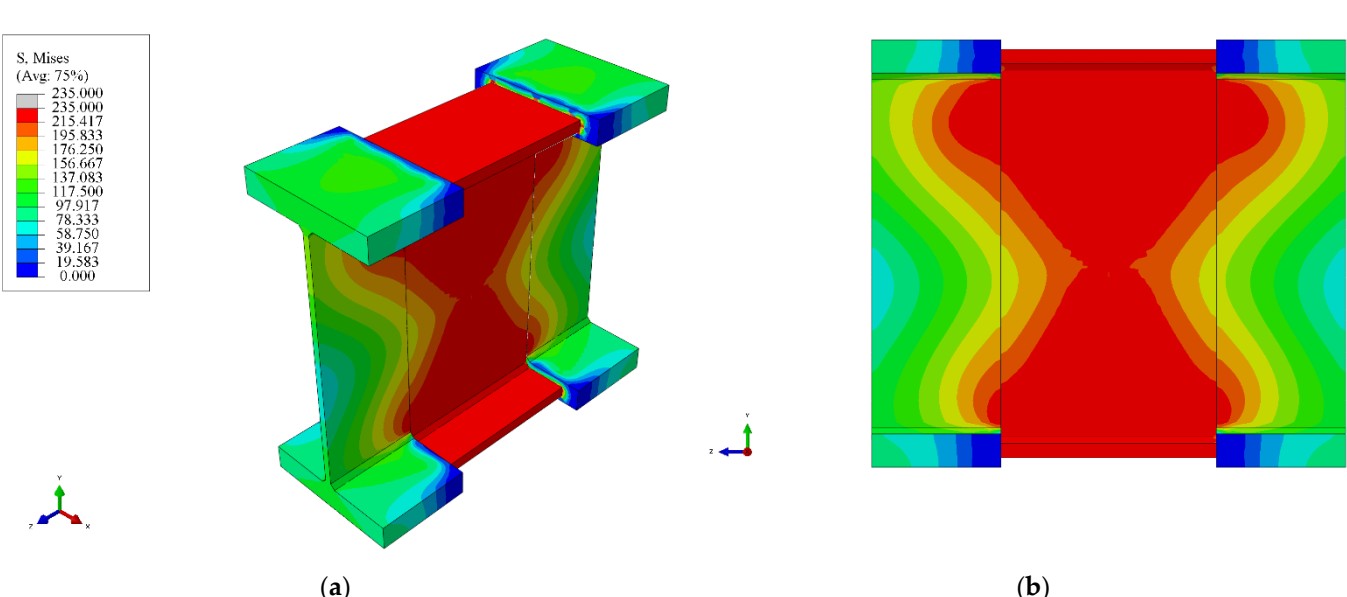

(**a**)                                                                                          (**b**)

**Figure 15.** von Mises' stress map (MPa) for LRPD IPE400 (DP1) at the ultimate loading condition: (**a**) perspective view; (**b**) lateral view.

In particular, the LRPD for IPE400 (DP1), Figure 15, shows a perfect plastic behavior strictly concentrated just in correspondence of the inner portion while the close outer portions behave in a substantially elastic way. This very good behavior is related to the special load condition characterized by a high bending moment and small axial force.

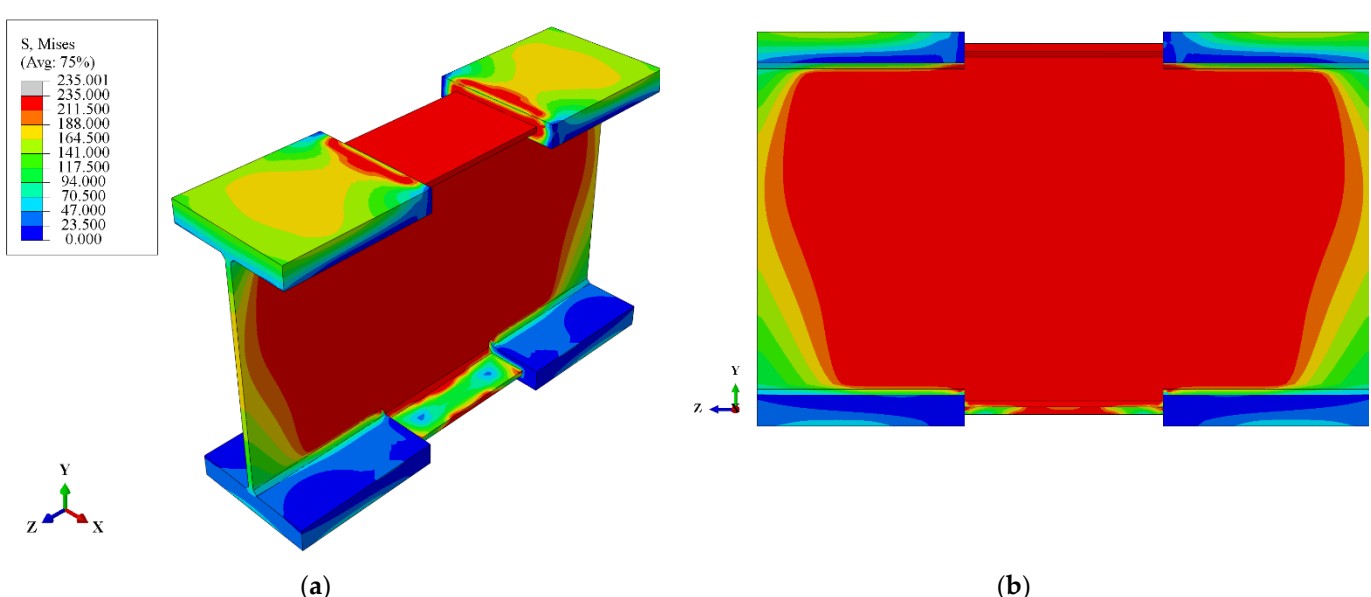

(**a**)                                                                                                                 (**b**)

**Figure 16.** von Mises' stress map (MPa) for LRPD IPE400 (DP2) at the ultimate loading condition: (**a**) perspective view; (**b**) lateral view.

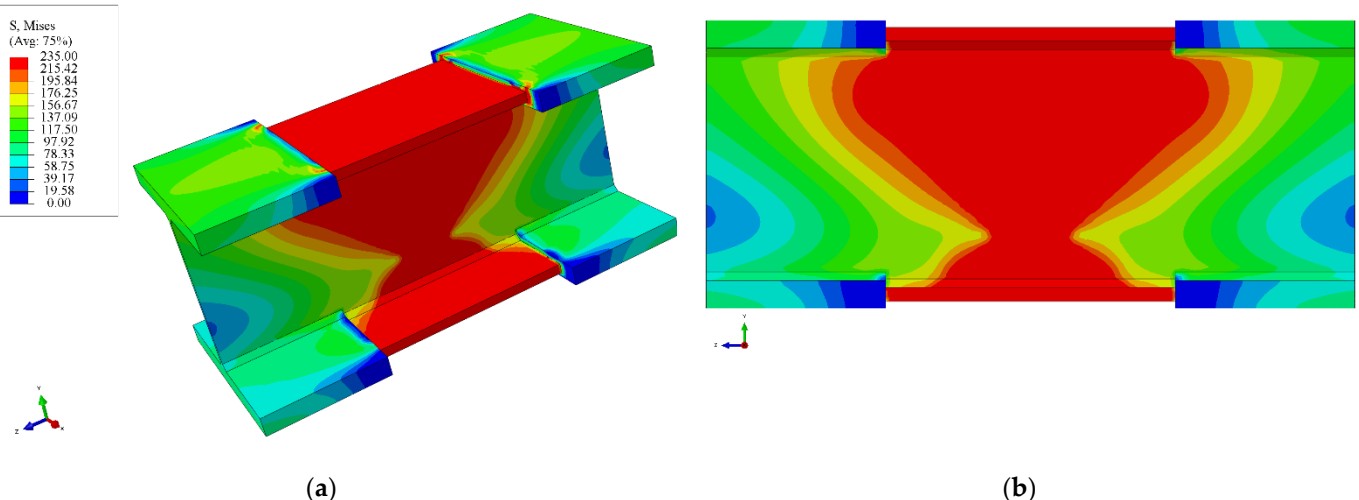

(**a**)                                                                                                                 (**b**)

**Figure 17.** von Mises' stress map (MPa) for LRPD HEA300 (DP1) at the ultimate loading condition: (**a**) perspective view; (**b**) lateral view.

Even the LRPD for IPE400 (DP2), Figure 16, shows the full plastic behavior of the inner portion but the close outer portions are strongly interested by the spread of the plastic deformations. However, it is worth noting that the flanges remain in elastic regime so that the elastic response to the acting moment is ensured. This particular behavior is related to the special load condition characterized by a high axial force and small bending moment.

The LRPD for HEA300 (DP1), Figure 17, and the LRPD for HEA300 (DP2), Figure 18, show substantially the same behavior that has been previously commented upon, with the difference being that the outer portions are less involved in plastic deformation due to the greater length of the inner portion.

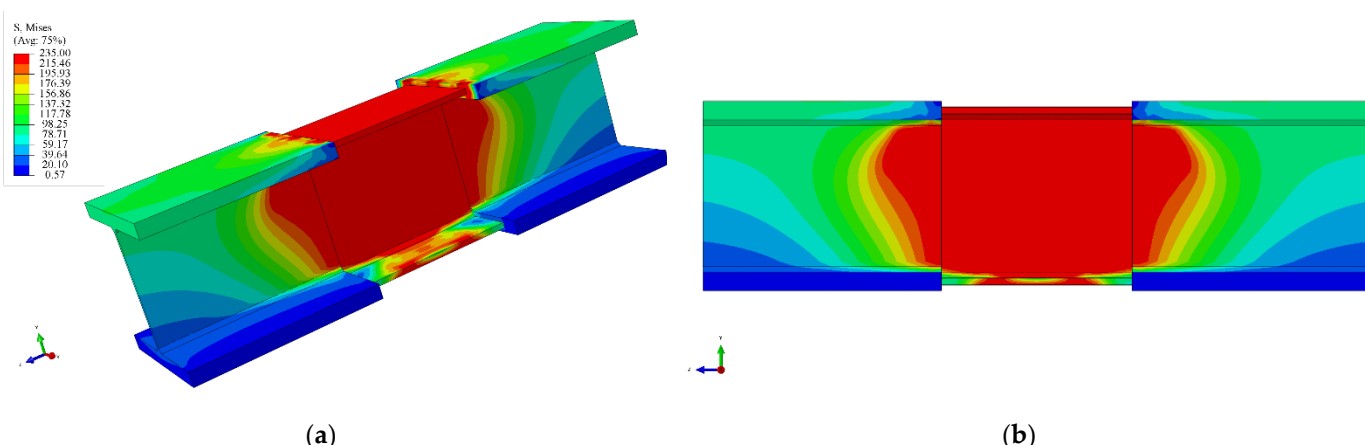

**(a)**  **(b)**

**Figure 18.** von Mises' stress map (MPa) for LRPD HEA300 (DP2) at the ultimate loading condition: (**a**) perspective view; (**b**) lateral view.

Since the LRPD is thought as a device connecting structural elements subjected to cyclic loadings, such as those arising during earthquakes, it is appropriate to evaluate the cyclic behavior of the designed device in terms of bending moment vs. bending curvature. This evaluation is always performed in the ABAQUS environment, referring to the device designed for the IPE400 profile (design point 1), whose model is sketched in Figure 19a. As it has been performed in the case of the FEM analyses reported above, the model is represented by a cantilever beam with a length of $3\ell$, plus the thickness of the plate, clamped in section $A$ and with a pressure loading applied at free section $D$ whose intensity varies only along the $y$-axis. The resultant of the loading is a pure bending moment with the moment axis coincident with the $x$ geometrical axis. The maximum value of the resultant bending moment lies in the range $(-M_a = -215 \text{ kNm}, M_a = 215 \text{ kNm})$, and it constitutes one loading cycle. The reported range is subdivided into 100 steps for the subsequent evaluation of the curvature of the device. This curvature is obtained as the ratio between $\Delta\varphi_{BC}$ and the length $\ell$ of the device, where $\Delta\varphi_{BC}$ is the relative rotation between sections $B$ and $C$. In Figure 19b the results in terms of bending moment vs. bending curvature are reported compared with the corresponding one obtained in the case of the IPE400 cantilever beam without the LRPD with the same length and loading. As can be observed, the device, after just one cycle, shows a stationary behaviour (plastic shakedown) and, therefore, no further loading cycles have been performed since the results will be superimposable with those already obtained. In Figure 19b the diagram of the IPE400 cantilever beam shows, as was expected $(M_a < M_{pl})$, a linear behaviour due to the linear elastic behaviour of the beam. Further, the analysis of Figure 19b clearly shows that the proposed device perfectly carries out the role of receiver of plastic deformations; it also develops a satisfactory plastic dissipation showing an elastic behaviour coincident with that of the original profile. Analogous analysis has been performed in the case of HEA300 LRPD (design point 1), whose model and results are reported in Figure 20. The examination of these figures leads to analogous remarks, such as those reported above for the IPE400 LRPD and, therefore, for brevity's sake they are not reported.

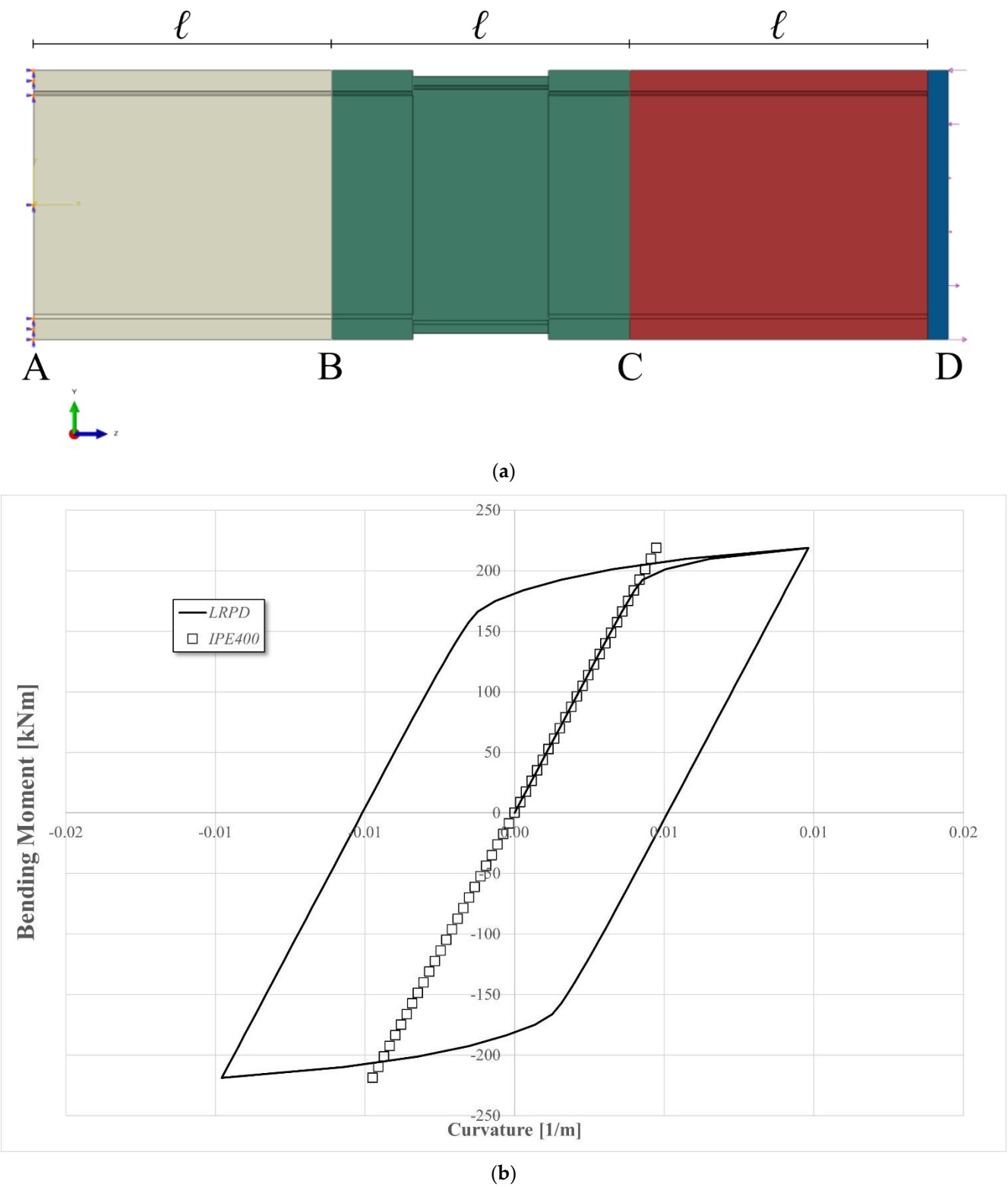

**Figure 19.** Cyclic analysis for IPE400 LRPD (DP1): (**a**) lateral view of the model; (**b**) bending moment vs. bending curvature curve.

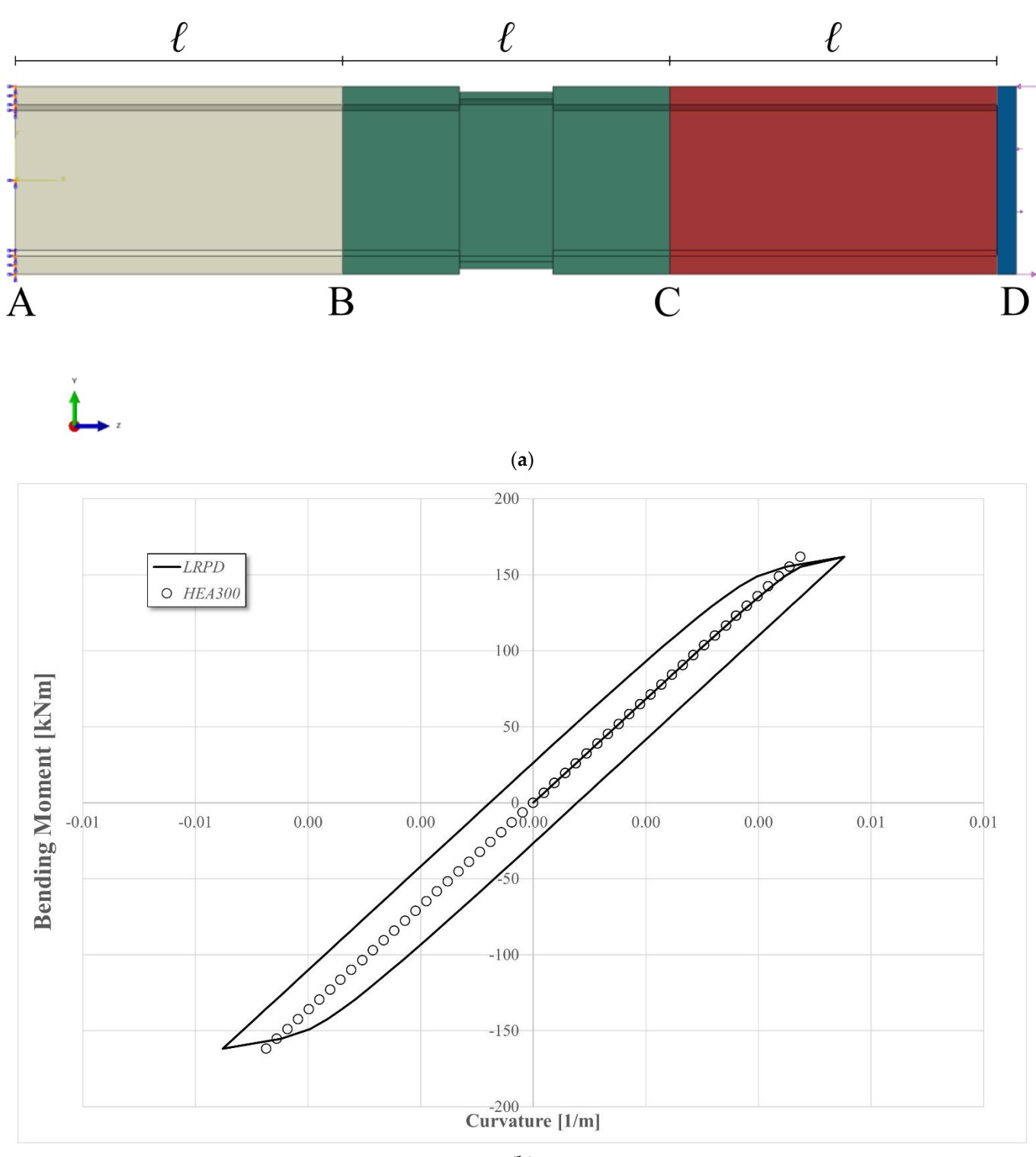

**Figure 20.** Cyclic analysis for HEA300 LRPD (DP1): (**a**) lateral view of the model; (**b**) bending moment vs. curvature curve.

## 5. Conclusions

In the present paper a new formulation of the optimal design problem devoted to obtaining the minimum volume of a special moment resisting connection device for steel elements has been proposed. The fundamental novelty is related to the introduction of suitable constraints which ensure the capability of the device to exhibit a full plastic

curvature in correspondence to the prefixed structural portion, avoiding any local and/or global buckling phenomenon.

In the application stage reference has been made to the design of devices for the most used steel profiles, i.e., HEA and IPE elements; for each chosen profile two different mechanical conditions are considered, so as to evaluate the sensibility of the chosen numerical procedure to the presence of the axial force and of the bending moment.

The obtained results allow us to observe:

1. the "fmincon" solver implemented by utilizing the MATLAB Optimization Toolbox showed good stability and reliability;
2. all of the obtained optimal designs of the devices exhibit the expected features imposed by the problem constraints. First of all, the full plastic curvature, even if for HEA profile and IPE profile some different positions must be fixed: $\beta = 1$ for HEA and $\beta = 0.5$ for IPE. These values are recommended in practical application, due to the different web flexural contribution of the referenced profiles;
3. the correctness of the expected behavior of the device has been verified by performing suitable FEM analyses in ABAQUS environment;
4. further studies are expected, for example regarding a campaign of experimental analysis, the design of appropriate steel connection with the beam element, the study of the elastic and plastic behavior of steel frame equipped with the proposed devices, and the utilization of steel frame equipped with the devices in the restoration of masonry structure building.

**Author Contributions:** Conceptualization, L.P. and S.B.; methodology, L.P., S.B. and S.V.; software, S.V.; validation, L.P. and S.B.; writing—original draft preparation, L.P. and S.B.; writing—review and editing, L.P. and S.B.; supervision, L.P. and S.B. All authors have read and agreed to the published version of the manuscript.

**Funding:** This research received no external funding.

**Institutional Review Board Statement:** Not applicable.

**Informed Consent Statement:** Not applicable.

**Conflicts of Interest:** The authors declare no conflict of interest.

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
