# Peer review of "A New Design Problem in the Formulation of a Special Moment Resisting Connection Device for Preventing Local Buckling"

_applsci, doi:10.3390/app12010202_

Round 1

Reviewer 1 Report

The paper presents a parametric study on the optimal design of the proposed LRPD in terms of the new approach. It is an interesting topic to discuss the problem from a structural point of view. However, it is believed, the paper needs to detail in various sections of the article body in terms of the research discussion and concept clarity. The Review comments are included in the manuscript as uploaded, annotated, PDF copy of this paper. It is suggested to provide responses and updates for the comments in a separate file with a clear explanation and location of the update in the manuscript.

Author Response

Authors are grateful to the reviewers for their suggestions and remarks. The paper has been deeply revised and improved following the remarks of the reviewers. In the attached file a point by point reply is reported.

Reviewer 2 Report

Find in the attached file

Author Response

Authors are grateful to the reviewer for his suggestions and remarks. The paper has been deeply revised and improved following the remarks of the reviewers. In the attached file a point by point reply is reported.

Reviewer 3 Report

There is no Methods part in the article.

(1), (2), (3) equations are repeated twice.

Explanations of the equations should begin with "where".

Figure 5 is not detalized - what is presented in parts a), b), c).

Rounding of numbers should be equalized in Table 3.

Author Response

Authors are grateful to the reviewer for his suggestions and remarks. The paper has been deeply revised and improved following the remarks of the reviewers. In the following a point by point reply is reported.

Round 2

Reviewer 1 Report

The reviewer responded to all my comments. Therefore, it is accepted in present form. 

Reviewer 2 Report

 No further suggestions and comments.